# Dehydration and insulinopenia are necessary and sufficient for euglycemic ketoacidosis in SGLT2 inhibitor-treated rats

Rachel J. Perry[1,2], Aviva Rabin-Court[1], Joongyu D. Song[1], Rebecca L. Cardone[1], Yongliang Wang[1], Richard G. Kibbey[1,2] & Gerald I. Shulman [1,2]

Sodium-glucose transport protein 2 (SGLT2) inhibitors are a class of anti-diabetic agents; however, concerns have been raised about their potential to induce euglycemic ketoacidosis and to increase both glucose production and glucagon secretion. The mechanisms behind these alterations are unknown. Here we show that the SGLT2 inhibitor (SGLT2i) dapagliflozin promotes ketoacidosis in both healthy and type 2 diabetic rats in the setting of insulinopenia through increased plasma catecholamine and corticosterone concentrations secondary to volume depletion. These derangements increase white adipose tissue (WAT) lipolysis and hepatic acetyl-CoA content, rates of hepatic glucose production, and hepatic ketogenesis. Treatment with a loop diuretic, furosemide, under insulinopenic conditions replicates the effect of dapagliflozin and causes ketoacidosis. Furthermore, the effects of SGLT2 inhibition to promote ketoacidosis are independent from hyperglucagonemia. Taken together these data in rats identify the combination of insulinopenia and dehydration as a potential target to prevent euglycemic ketoacidosis associated with SGLT2i.

[1] Departments of Internal Medicine, Yale University School of Medicine, P.O. Box 208020TAC S269, New Haven, CT 06519, USA. [2] Departments of Cellular and Molecular Physiology, Yale University School of Medicine, P.O. Box 208020TAC S269, New Haven, CT 06519, USA. Correspondence and requests for materials should be addressed to G.I.S. (email: gerald.shulman@yale.edu)

SGLT2 inhibitors are effective glucose-lowering agents due to their ability to promote glycosuria[1–8]. However, concerns have been raised that they might promote euglycemic ketoacidosis[9–20], a potentially fatal condition. Euglycemic ketoacidosis is rare in type 2 diabetic patients, with incidence of ~0.5% (~5 cases per 1000 person-years)[9,21,22]. However, in type 1 diabetic patients, euglycemic ketoacidosis has higher incidence (6 to 20%, or 60–200 cases per 1000 person-years)[23,24]. Thus, understanding the mechanism by which SGLT2 inhibitors can provoke euglycemic ketoacidosis and increase hepatic glucose production would be of great clinical benefit in determining whether there are steps patients can take upon initiation of the drug to reduce these risks. Several potential mechanisms have been proposed for euglycemic ketoacidosis associated with SGLT2i, including reductions in pancreatic β-cell secretion of insulin[25–28] and increased plasma glucagon concentrations due to direct pancreatic α-cell stimulation[29–31]. As insulin is a potent suppressor of WAT lipolysis and hepatic ketogenesis, insulinopenia per se could explain part or possibly all of the ketoacidosis observed with SGLT2 inhibition, particularly in combination with increased lipid oxidation as has been observed in humans[32,33] and rodents[34,35]. Increases in plasma glucagon concentrations have been directly attributed to reduced α-cell SGLT2-mediated glucose transport[29,31], though the rationale for this mechanism has been debated[36]. Reduced paracrine signaling by insulin due to the glucose-lowering effect of SGTL2 inhibition has also been suggested to be the major factor responsible for the observed increases in plasma glucagon, hepatic glucose production, and ketogenesis[27,28,30,37]. It has also been proposed that SGLT2-inhibition increases plasma ketone concentrations through a direct effect on the kidney by promoting renal reabsorption of acetoacetate[38]. However a recent study found that renal β-hydroxybutyrate (β-OHB) clearance increased modestly after treatment with the SGLT2i empagliflozin but represented less than 1% of the filtered load of β-OHB[22], suggesting that alterations in β-OHB clearance are unlikely to contribute much-if at all-to ketosis in those treated with an SGLT2 inhibitor. Taken together, the previously available data on ketoacidosis associated with SGLT2i do not provide a unifying mechanism and leave open three key questions regarding SGLT2i effects on in vivo metabolism: (1) what is the mechanism by which SGLT2 inhibition causes hyperglucagonemia?, (2) does this hyperglucagonemia contribute to euglycemic ketoacidosis and/or increased hepatic glucose production, and (3) if hyperglucagonemia is not sufficient to promote euglycemic ketoacidosis and increased hepatic glucose production following treatment with SGLT2i, what is the mechanism by which SGLT2 inhibitors promote euglycemic ketoacidosis?

To answer these questions, in this study we apply stable isotope tracer methods to assess in vivo rates of hepatic ketogenesis, white adipocyte (WAT) lipolysis, and hepatic glucose production following acute dapagliflozin treatment. Here we show that SGLT2i-induced euglycemic ketoacidosis requires both insulinopenia, as well as increases in plasma corticosterone and catecholamine concentrations secondary to volume depletion, which together lead to increased rates of WAT lipolysis, hepatic acetyl-CoA content, and hepatic ketogenesis. Additionally, we show using rat and human islets that, contrary to prior studies, dapagliflozin does not promote hyperglucagonemia through a direct effect on the pancreatic α-cell. We go on to show that SGLTi-induced glucagon secretion may be mediated at least in part through an autonomic nervous system response, and that this effect is not sufficient to cause ketoacidosis or increased hepatic glucose production.

## Results

**SGLT2 inhibition causes ketoacidosis in healthy rats.** In order to identify the mechanism by which SGLT2 inhibition can cause euglycemic ketoacidosis, we treated normal Sprague-Dawley (SD) rats with dapagliflozin (10 mg kg$^{-1}$) and sacrificed them six hours after treatment, after fasting for a total of eight hours. Administering dapagliflozin led to pronounced glycosuria associated with a ~25 mg dL$^{-1}$ reduction in plasma glucose concentrations as compared to vehicle-treated rats six hours after treatment (Fig. 1a, Supplementary Fig. 1a). Dapagliflozin-treated rats, which had their drinking water withheld throughout the 6 h period following dapagliflozin treatment, were ketoacidotic, exhibiting an eight-fold increase in plasma β-hydroxybutyrate (β-OHB) concentrations, a fifteen-fold increase in urine β-OHB concentrations, a 2.5-fold increase in plasma acetoacetate concentrations and a 30% reduction in plasma bicarbonate concentrations, reflecting a four-fold increase in whole-body β-OHB turnover and a 50% reduction in β-OHB clearance (Fig. 1b–e, Supplementary Fig. 1b, c). This increase in hepatic ketogenesis was associated with 2–3 fold increases in WAT lipolysis, as well as 85% reductions in liver concentrations of the CPT1 inhibitor malonyl-CoA[39,40] (Fig. 1f, Supplementary Fig. 1d–g). As predicted by their lower plasma glucose concentrations, dapagliflozin-treated rats exhibited a 50% reduction in plasma insulin concentrations despite a 75% increase in hepatic acetyl-CoA content, which could be attributed to dapagliflozin-induced WAT lipolysis and which was associated with a similar increase in rates of endogenous glucose production (Fig. 1g, h, Supplementary Fig. 1h).

Next we sought to determine the mechanism of the observed increases in hepatic ketogenesis with SGLT2 inhibition. Not surprisingly, we observed a 2.7-fold increase in volume loss over the 6 h duration of the study in dapagliflozin-treated rats not given free access to water relative to controls, which caused extracellular volume depletion as reflected by increases in plasma angiotensin-II and antidiuretic hormone (ADH) concentrations (Fig. 2a, Supplementary Fig. 2a, b). This volume depletion was associated with increases in plasma catecholamine concentrations and HPA axis activity as reflected by increases in plasma ACTH and corticosterone concentrations (Fig. 2b, c, Supplementary Fig. 2c, d). We also observed a modest suppression of plasma leptin concentrations (Supplementary Fig. 2e), which can be attributed to the reductions in plasma glucose and insulin concentrations[41], and may have also contributed to the increase in plasma corticosterone concentrations[42–44]. In addition, and consistent with prior studies, SGLT2 inhibitor treatment was associated with a marked increase in plasma glucagon concentrations but did not alter plasma growth hormone concentrations (Fig. 2d, Supplementary Fig. 2f).

To examine whether SGLT2i-induced volume depletion was the cause of the increased WAT lipolysis and catecholamine/corticosterone concentrations, we treated a group of rats with dapagliflozin and either infused saline to match end-study body weights to what was measured in control animals, or gave rats free access to water throughout the 6 h experiment (Fig. 2a). These interventions did not alter plasma or urine glucose concentrations relative to dapagliflozin-treated rats, but reduced activity of the renin-angiotensin-aldosterone (RAA) axis, HPA axis, and plasma catecholamine concentrations to control levels. Most importantly, the normalization of the plasma catecholamine and corticosterone concentrations was associated with reductions in rates of WAT lipolysis, hepatic acetyl-CoA content, and hepatic ketogenesis to levels measured in untreated control rats (Figs. 1, 2, Supplementary Fig. 1, 2). Surprisingly these changes

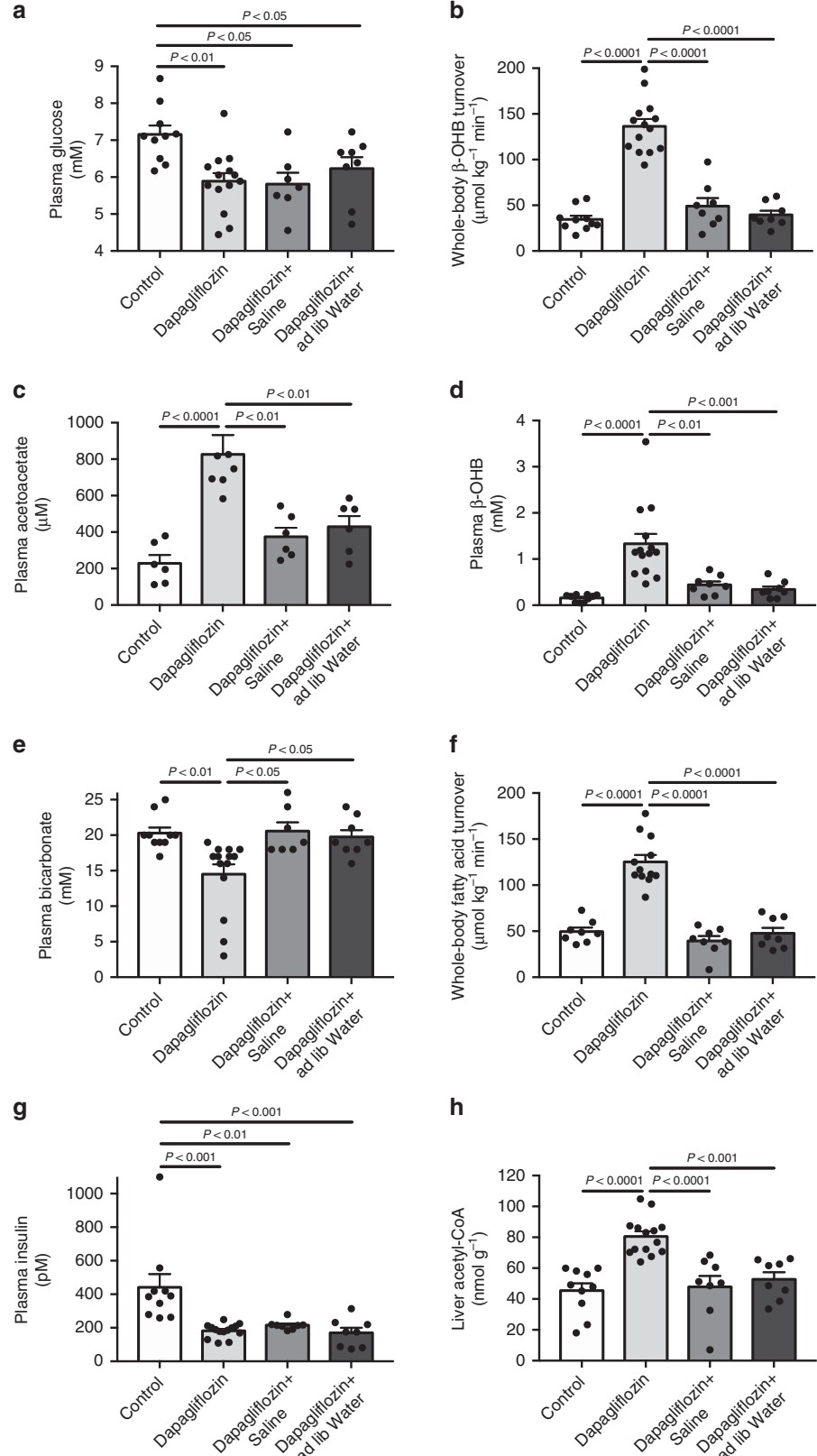

**Fig. 1** Dapagliflozin causes increases in rates of hepatic ketogenesis and glucose production due to extracellular volume depletion. **a–d** Plasma glucose, bicarbonate, acetoacetate, and β-OHB concentrations. In panel **c**, $n = 7$ dapagliflozin-treated and $n = 6$ in all other groups. **e**, **f** Whole-body β-OHB and fatty acid turnover. **g** Plasma insulin concentrations. **h** Liver acetyl-CoA. Data are the mean ± S.E.M., and groups were compared by ANOVA with Bonferroni's multiple comparisons test

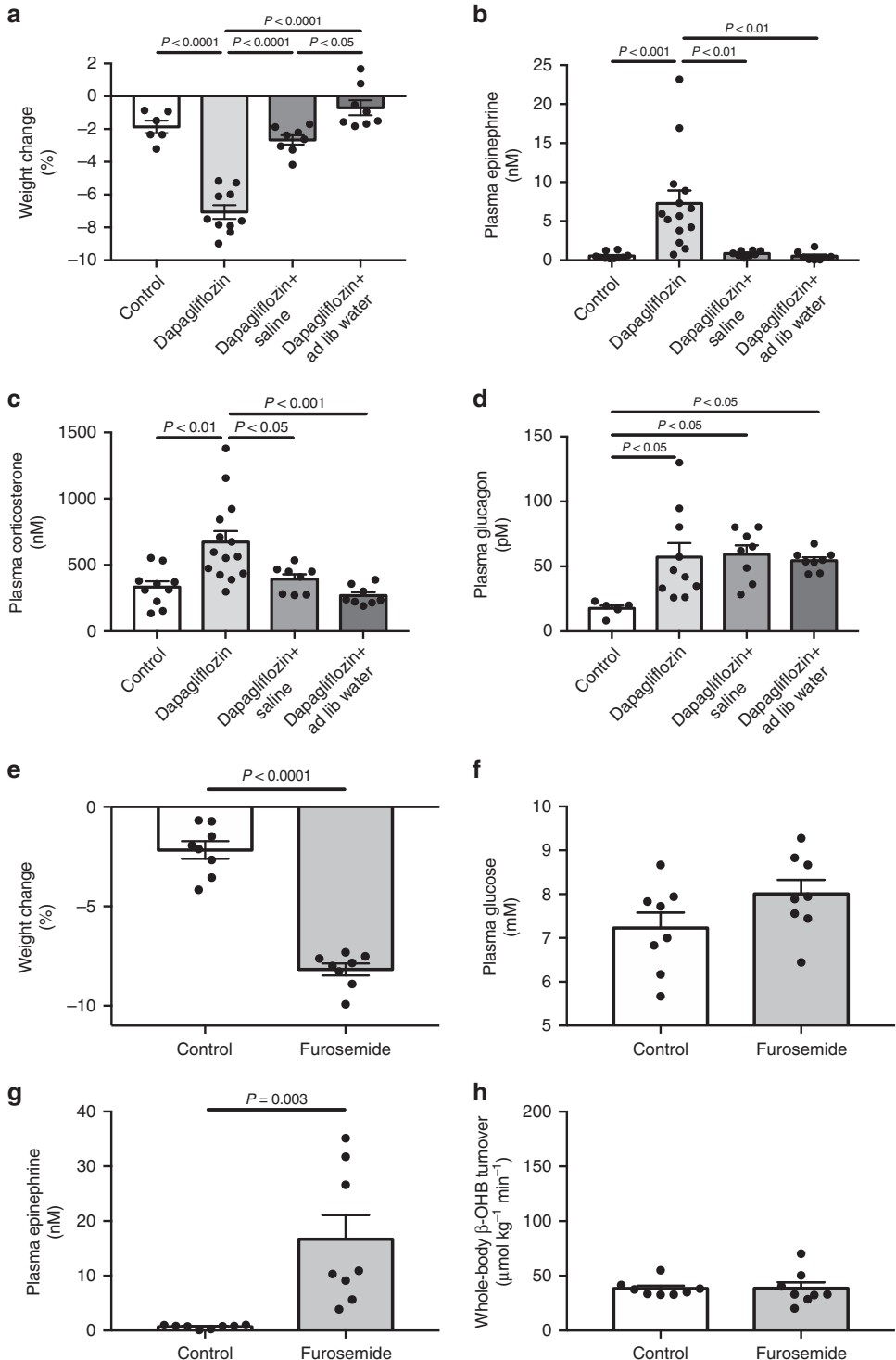

**Fig. 2** Dapagliflozin-induced increases in hepatic ketogenesis and hepatic glucose production are caused by extracellular volume depletion. **a** Weight change from baseline (time of injection) measured six hours after treatment. **b**–**d** Plasma epinephrine, corticosterone, and glucagon concentrations. Data are the mean ± S.E.M., with comparisons by ANOVA with Bonferroni's multiple comparisons test. **e**, **f** Weight change and plasma glucose concentrations six hours after treatment with furosemide. **g** Plasma epinephrine. **h** Whole-body β-OHB turnover. In panels **e**–**h**, comparisons were performed using the 2-tailed unpaired Student's *t*-test. Data are the mean ± S.E.M

occurred without any differences in plasma insulin or glucagon concentrations or hepatic malonyl-CoA concentrations relative to dapagliflozin-treated rats.

**Dehydration is necessary but not sufficient for ketoacidosis**. To examine whether dehydration is sufficient to cause euglycemic

ketoacidosis in rats, we treated healthy rats with furosemide (40 mg kg$^{-1}$) and measured rates of WAT lipolysis and ketogenesis six hours later, following an 8 h fast. Furosemide treatment led to extracellular volume depletion and dehydration, indicated by increases in RAA activity, similar to that observed in dapagliflozin-treated rats (Fig. 2e, Supplementary Fig. 3a, b). This

dehydration was associated with increases in both plasma corticosterone and catecholamine concentrations; however, furosemide did not alter plasma glucose, insulin, glucagon, or growth hormone concentrations (Fig. 2f, g, Supplementary Fig. 3c–g) in these recently fed animals. Similarly, we observed no differences in rates of WAT lipolysis, ketogenesis, acetyl-CoA or malonyl-CoA, or endogenous glucose production, all of which were similar to those measured in untreated controls (Fig. 2h, Supplementary 3h–p). These data suggest that an additional perturbation is required together with volume depletion-induced catecholamine/glucocorticoid release to increase hepatic ketogenesis in this model.

**Insulinopenia is necessary but not sufficient for ketoacidosis.** We hypothesized that this additional factor could be reductions in plasma insulin concentrations. To test this hypothesis, we measured rates of WAT lipolysis and ketogenesis in dapagliflozin-treated rats before and after infusion of glucose to raise plasma glucose and insulin concentrations to the concentrations measured in control rats: ~6.7 mM and ~400 pM, respectively (Supplementary Fig. 4a, b). This intervention abrogated dapagliflozin-induced increases in hepatic ketogenesis and hepatic glucose production by suppressing WAT lipolysis (Supplementary Fig. 4c–f). As predicted by the WAT lipolysis data, glucose infusion lowered hepatic acetyl-CoA concentrations in dapagliflozin-treated rats to concentrations similar to those observed in control rats ($65 \pm 2$ nmol g$^{-1}$, $p = 0.01$ versus dapagliflozin-treated rats, Fig. 1h) without affecting hepatic malonyl-CoA content relative to dapagliflozin-treated rats ($0.34 \pm 0.11$ nmol g$^{-1}$) and reduced endogenous glucose turnover by ~50%, normalizing glucose turnover to rates similar to those measured in controls (Supplementary Fig. 4g). Insulin's effect to suppress WAT lipolysis, ketogenesis, and endogenous glucose production occurred in the absence of any effect on plasma catecholamine, corticosterone, or glucagon concentrations (Supplementary Fig. 4h–k).

**Dehydration with insulinopenia is sufficient for ketoacidosis.** In order to examine whether the combination of dehydration and insulinopenia is sufficient to promote euglycemic ketoacidosis in the rat, we treated 48 h fasted, insulinopenic rats with furosemide to provoke dehydration (Supplementary Fig. 5a). In insulinopenic rats, furosemide treatment caused a modest increase in plasma glucose and insulin concentrations associated with 100–600% increases in plasma catecholamine and glucocorticoid concentrations, with this increase in sympathetic activity reflected in an 40% increase in heart rate (Supplementary Fig. 5b–f). Similar to recently fed dapagliflozin-treated rats, insulinopenic furosemide-treated rats exhibited 70% increases in WAT lipolysis and ketogenesis leading to euglycemic ketoacidosis (Supplementary Fig. 5g–q). Taken together, these data demonstrate that dehydration leading to increases in lipolytic hormone concentrations, and simultaneous insulinopenia, are both necessary and sufficient to cause euglycemic ketoacidosis in normal awake SD rats.

**Dapagliflozin does not cause hyperglucagonemia directly.** Next we sought to understand the mechanism by which dapagliflozin promotes hyperglucaonemia in these healthy rats. We found no direct effect of either dapagliflozin or canagliflozin to alter insulin or glucagon secretion in isolated rat (Supplementary Fig. 6a, b) or human islets (Supplementary Fig. 6c–h) across a wide range of glucose concentrations and in two commonly used buffers. We next hypothesized that hyperglucagonemia may occur through a central mechanism in dapagliflozin-treated rats. To test this hypothesis, we performed an intracerebroventricular (ICV) injection of a low dose of dapagliflozin (4.69 μg kg$^{-1}$) which did

not reach the peripheral circulation, as evidenced by the absence of any diuretic effect or glycosuria (Supplementary Fig. 7a, b). However, this central injection of dapagliflozin caused hyperglucagonemia, increasing plasma glucagon concentrations threefold and increasing plasma glucose, insulin, and endogenous glucose production without any alterations in plasma catecholamines, WAT lipolysis or ketogenesis (Supplementary Fig. 7c–m). To further demonstrate the ability of dapagliflozin to promote hyperglucagonemia through CNS mechanisms, we injected dapagliflozin in normal rats following a left-sided vagotomy and found that unilateral vagotomy significantly blunted dapagliflozin's effect to promote hyperglucagonemia despite causing dehydration similar to that seen in dapagliflozin-injected controls and causing ketoacidosis (Supplementary Fig. 8a–d).

**SGLT2 inhibition causes ketoacidosis in a rat model of type 2 diabetes.** Next we verified the applicability of these findings in a rat model of T2D in which β-cell function is limited by low-dose streptozotocin and nicotinamide in combination with insulin resistance generated by high fat feeding[45]. In T2D rats not given access to drinking water, a pharmacologically relevant dose of dapagliflozin (1 mg kg$^{-1}$) caused glycosuria and dehydration, which resulted in increases in plasma corticosterone and catecholamine concentrations and mean arterial pressure despite a marked effect to lower plasma glucose and insulin concentrations (Fig. 3a–e, Supplementary Fig. 9a–f). The observed increases in catecholamine and glucocorticoid concentrations were associated with increased white adipose tissue lipolysis, resulting in diabetic ketoacidosis (Fig. 3f–h, Supplementary Fig. 9g–l). These findings speak to the potential of this mechanism to explain cases of diabetic ketoacidosis associated with SGLT2i in clinical practice.

**SGLT2 inhibition promotes lipolysis via glucocorticoid and β−1 adrenergic activity.** Finally, we investigated the mechanism by which SGLT2i-induced dehydration increased WAT lipolysis. Having observed increases in both plasma catecholamine and corticosterone concentrations, we treated dapagliflozin-treated rats with pharmacologic inhibitors of β$_1$-adrenergic and/or glucocorticoid receptor activity (atenolol and mifepristone, respectively). These studies demonstrate that, whereas only β$_1$-adrenergic activation increases heart rate and body temperature in response to dapagliflozin treatment, both glucocorticoid and β$_1$-adrenergic activity promote WAT lipolysis, β-oxidation, glucose production, and ketogenesis, with the combination of glucocorticoid and β$_1$-adrenergic blockade fully preventing dapagliflozin's ability to promote each of these derangements (Fig. 4a–i, Supplementary Fig. 10a–j). Interestingly β$_1$-adrenergic blockade also abrogated hyperglucagonemia in dapagliflozin-treated rats, indicating that SGLT2i-induced hyperglucagonemia may be at attributed at least in part to β$_1$-adrenergic activation secondary to dehydration at least in this rodent model.

### Discussion
Multiple mechanisms have been proposed to explain the pathogenesis of euglycemic ketosis in patients taking SGLT2 inhibitors, including hyperglucagonemia, insulinopenia, and reduced renal β-OHB clearance. However, because ketone turnover has never been measured in SGLT2i-treated subjects, prior to this study it had not been demonstrated whether euglycemic ketosis occurs due to increased ketone production, reduced clearance, or both. Here we show that the rate of whole-body β-OHB appearance is markedly increased following high-dose dapagliflozin treatment in both normal awake SD rats and an obese rat model of poorly-controlled T2D (Figs. 1b and 3g). These data beg the question of why hepatic ketogenesis is increased. To that end, we measured

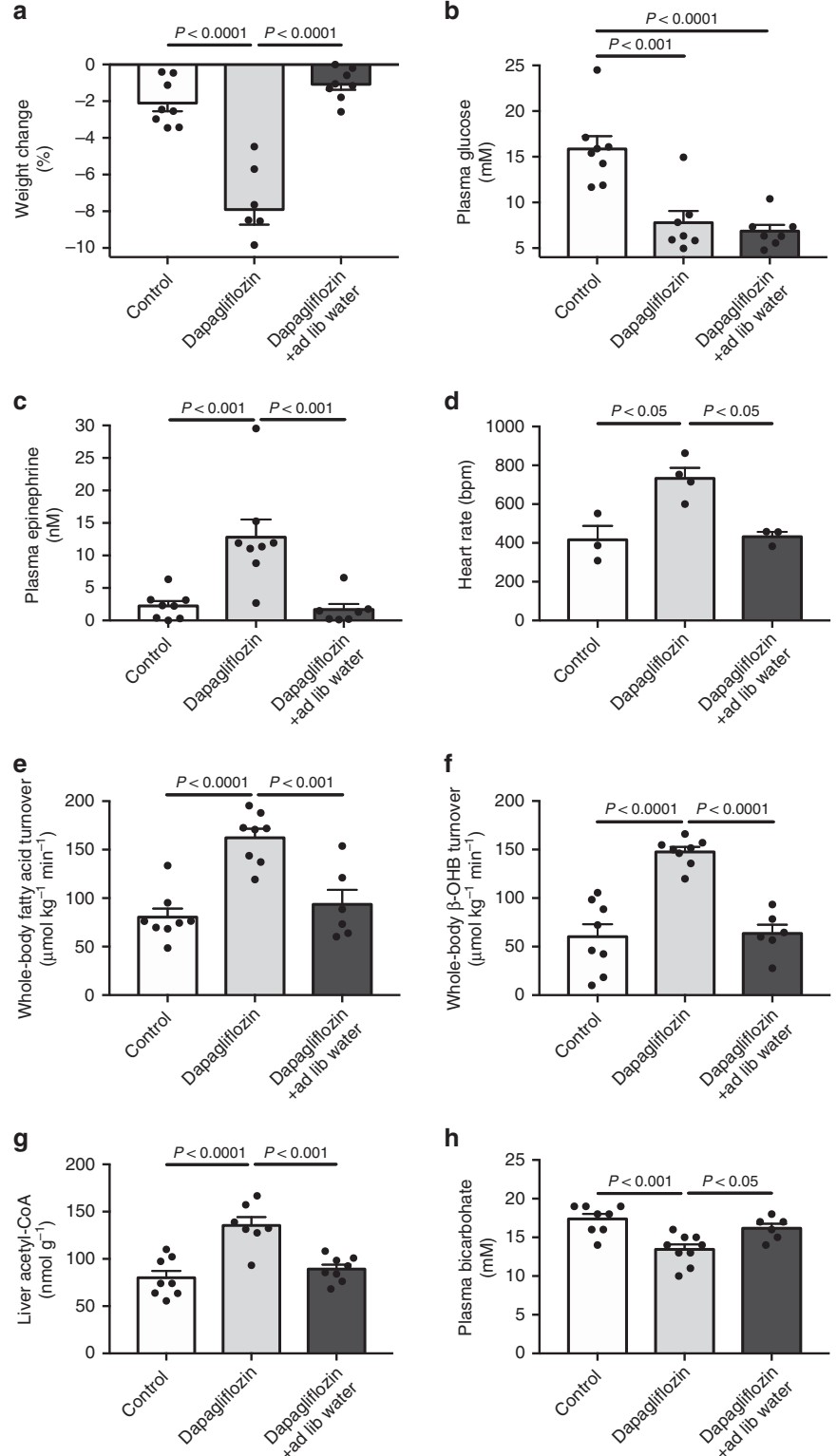

**Fig. 3** Dapagliflozin (1 mg kg$^{-1}$) causes ketoacidosis in a rat model of type 2 diabetes. **a** Weight change. **b**, **c** Plasma glucose and epinephrine concentrations. **d** Heart rate. **e**, **f** Whole-body fatty acid and β-OHB turnover. **g** Liver acetyl-CoA. **h** Plasma bicarbonate. In all panels, data are the mean ± S. E.M., with comparisons by ANOVA with Bonferroni's multiple comparisons test

rates of WAT lipolysis and observed 2–3 fold increases in rates of fatty acid and glycerol turnover, as well as hepatic acetyl-CoA content. These increases in WAT lipolysis were associated with increased HPA axis activity, as well as plasma catecholamine concentrations. Volume replacement by saline infusion or ad lib access to water throughout the study prevented these increases in plasma corticosterone and catecholamines, WAT lipolysis, hepatic acetyl-CoA content, and hepatic ketogenesis in both healthy control and T2D rats, demonstrating that volume depletion is necessary to provoke ketoacidosis with SGLT2i treatment. The requirement for volume depletion to generate ketoacidosis in SGLT2i treated rats could also explain why

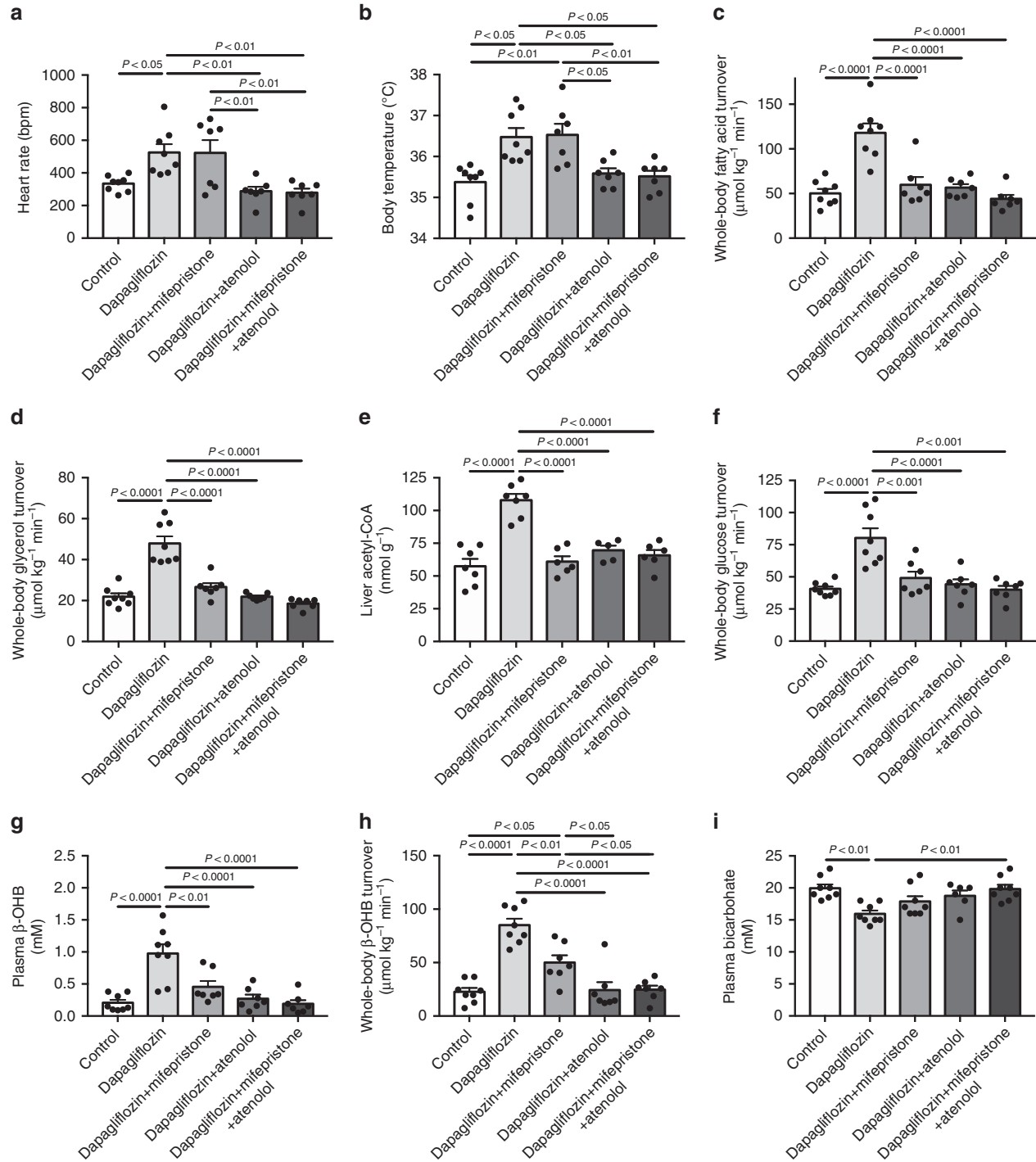

**Fig. 4** Dapagliflozin causes ketoacidosis in normal rats through β-1 adrenergic and glucocorticoid activity. **a**, **b** Heart rate and body temperature after treatment with dapagliflozin ± atenolol ± mifepristone. **c**, **d** Whole-body fatty acid and glycerol turnover. **e** Liver acetyl-CoA content. **f** Whole-body glucose turnover. **g**, **h** Plasma β-OHB concentrations and β-OHB turnover. **i** Plasma bicarbonate. In all panels, data are the mean ± S.E.M., with comparisons by ANOVA with Bonferroni's multiple comparisons test

ketoacidosis associated with SGLT2i is relatively rare in humans: given ad lib access to water, most individuals will not become dehydrated if their thirst center is intact. However in cases such as illness, reduced food/water intake, or alcohol intake, which can all promote dehydration and which have all been identified as factors predisposing to ketoacidosis[10], can predispose patients taking an SGLT2i to develop euglycemic ketoacidosis.

In this regard data demonstrating that volume depletion resulting from dapagliflozin treatment causes increases in plasma

catecholamines and HPA axis activity beg the question of why commonly-used loop diuretics capable of promoting similar reductions in extracellular volume have not also been reported to cause ketoacidosis. To examine this question, we treated recently fed, awake, SD rats with high-dose furosemide (40 mg kg⁻¹) and measured rates of WAT lipolysis and ketogenesis six hours later, following an 8 h fast. In contrast to dapagliflozin, furosemide did not alter WAT lipolysis, hepatic acetyl-CoA, or hepatic ketogenesis, despite increasing plasma catecholamine and

corticosterone concentrations. These data suggest that an additional perturbation is required in addition to volume depletion-induced increases in catecholamine/glucocorticoid concentrations to permit increased ketogenesis in this model. We hypothesized that this additional factor could be reductions in plasma insulin concentrations, as insulin is well known to be an acute suppressor of WAT lipolysis[46–49]. Consistent with this hypothesis, we found that furosemide-induced acute dehydration in otherwise healthy insulinopenic 48 h starved rats caused ketoacidosis and that infusion of glucose to raise plasma glucose concentrations to ~6.7 mM, similar to plasma glucose concentrations in untreated controls, suppressed WAT lipolysis and β-OHB turnover to levels measured in control rats. These data, consistent with our previous study[42] confirm that modest increases in plasma insulin concentrations (from ~175 to ~400 pM) can abrogate the effects of increased plasma catecholamines and corticosterone to stimulate WAT lipolysis and ketogenesis in dapagliflozin-treated rats and that volume depletion can provoke ketoacidosis only when plasma insulin concentrations are reduced below a critical threshold. It is, of course, possible that the relative importance of volume depletion versus insulinopenia in driving ketoacidosis associated with SGLT2i may differ between rodents and humans.

Consistent with prior studies[27,29–31,50–52], we found that plasma glucagon concentrations increased (Fig. 2d) with dapagliflozin treatment. However, we demonstrate that these changes can be dissociated from the development of ketoacidosis observed in SGLT2i-treated rats. Normalizing rats' extracellular volume status by infusion of saline or ad lib access to water prevented ketoacidosis without altering plasma insulin or glucagon concentrations or hepatic malonyl-CoA content relative to dapagliflozin-treated rats, but reduced rates of WAT lipolysis, β-OHB turnover, and hepatic acetyl-CoA content. These data demonstrate that a decreased portal insulin:glucagon ratio per se is not the sole mechanism for SGLT2 inhibitor-induced increases in hepatic ketogenesis and hepatic glucose production. Next, we investigated the mechanism by which dapagliflozin promotes hyperglucagonemia. Notably, whole body knockout of SGLT2 in mice on regular chow, high fat diet and db/db background did not impact glucagon secretion from isolated islets[53]. Surprisingly a subsequent study[29] showed that dapagliflozin promoted glucagon secretion from human islets. In the current study, we found that glucagon secretion was uniformly lower in islets incubated in Krebs-Ringer Modified Buffer (KRB) that lacks amino acids as compared to the more physiologic Dulbecco's Modified Eagle's Media (DMEM) confirming the importance of physiologic buffer selection when performing in vitro experiments; however, we did not observe any effect of either dapagliflozin or canagliflozin to promote glucagon secretion from isolated islets.

In the absence of a direct effect of SGLT2 inhibitors to directly promote glucagon secretion from isolated islets, and because SGLT2s are found in the brain[54], we hypothesized that dapagliflozin may promote glucagon secretion through a central mechanism. To test this hypothesis, we performed an intracerebroventricular (ICV) injection of a low dose of dapagliflozin (4.69 µg kg$^{-1}$) which did not reach the peripheral circulation, as evidenced by the absence of any diuretic effect or glycosuria (Supplementary Fig. 7a, b). Based on the absence of a peripheral effect of dapagliflozin, these data demonstrate that dapagliflozin may act through a central mechanism to promote hyperglucagonemia and that dapagliflozin stimulation of glucagon secretion can be dissociated from increases in hepatic ketogenesis or hepatic glucose production, particularly in the setting of hyperinsulinemia. Interestingly β$_1$-adrenergic inhibition, but not glucocorticoid receptor blockade, lowered plasma glucagon concentrations, consistent with previous reports suggesting an effect of catecholamines to promote α-cell glucagon release[55–57], and suggesting that the observed hyperglucagonemia in dapagliflozin-treated

rats, like the observed increases in WAT lipolysis and ketogenesis, may be partially catecholamine-dependent.

In summary, this study provides several new mechanistic insights into the pathogenesis of SGLT2i-induced euglycemic ketoacidosis, alterations in hepatic glucose production and glucagon secretion in rodents. Specifically we show that: (1) dapagliflozin promotes increased rates of hepatic ketogenesis and hepatic glucose production through a central mechanism resulting in stimulation of WAT lipolysis, (2) dapagliflozin promotes increased rates of WAT lipolysis by increasing both plasma catecholamine and corticosterone concentrations, as well as reducing plasma insulin concentrations which are both necessary for this process as demonstrated by the fact that glucose infusion abrogates the effect of dapagliflozin to promote WAT lipolysis, (3) dapagliflozin-induced increases in plasma glucocorticoid and β$_1$-adrenergic activity are secondary to extracellular volume depletion as evidenced by the prevention of these changes when extracellular volume depletion was avoided, (4) increases in plasma corticosterone and catecholamine concentrations secondary to extracellular volume depletion alone are not sufficient to induce increases in hepatic ketogenesis or hepatic glucose production in the absence of a concomitant reduction in plasma insulin concentrations as evidenced by the furosemide treatment studies, (5) dapagliflozin-induced hyperglucagonemia occurs by a central mechanism but this hyperglucagonemia does not mediate SGLT2i-induced increases in rates of hepatic ketogenesis and hepatic glucose production as reflected by the dissociation of hyperglucagonemia and the resolution of dapagliflozin-induced increases in hepatic ketogenesis and glucose production with volume repletion. Taken together this study demonstrates that two hits are required to provoke euglycemic ketoacidosis secondary to SGLT2i treatment in the rat, insulinopenia, as well as CNS activation of the HPA axis/increase catecholamine secretion. Further studies will be required to determine whether SGLT2i-induced dehydration is a potential target to prevent euglycemic ketoacidosis and increases in hepatic glucose production associated with SGLT2i in humans.

## Methods

**Animals**. Normal male Sprague-Dawley rats were obtained from Charles River weighing ~300 g. They were housed in the Yale Animal Resource Center on a 12 h light-dark cycle with ad lib access to regular chow (Envigo Teklad #2018, Franklin Township, NJ). One week prior to studies, rats underwent surgery under general isoflurane anesthesia to place catheters in the common carotid artery and internal jugular vein. A subgroup of rats underwent unilateral (left-sided) vagotomy, in which the vagus nerve was separated from the carotid artery by blunt dissection and cut with surgical scissors, a procedure which was immediately followed by placement of a catheter in the jugular vein. A separate group of rats were obtained from Charles River with catheters previously implanted in the 3$^{rd}$ ventricle. One week after arrival, vascular catheterization was performed at Yale as described above. To induce T2D, rats were fed high fat diet (60% calories from fat; Dyets #112245) for four weeks and underwent catheterization surgery in week 3. After an overnight fast, they were injected with 85 mg kg$^{-1}$ nicotinamide and, 10 min later, 40 mg kg$^{-1}$ streptozotocin and refed ad lib. They were studied two days later, with any rats with fed plasma insulin <125 pM, indicating type 1 diabetes, excluded from study. In all tracer studies, the arterial catheter was used for infusion of tracers and the venous catheter was used for blood sampling. All animal protocols were approved by the Yale University Institutional Animal Care and Use Committee, and all relevant ethical guidelines were followed.

**In vivo studies**. In most studies in this report, an intraperitoneal injection of 10 mg kg$^{-1}$ dapagliflozin (Sigma) or 40 mg kg$^{-1}$ furosemide (Sigma) in 10% DMSO/90% normal saline (total volume: 1 mL kg$^{-1}$), or an equivalent volume of 10% DMSO/90% saline vehicle, was administered to rats at 10 a.m. following two hours of food withdrawal. T2D rats were treated with an IP injection of 1 mg kg$^{-1}$ dapagliflozin (Astra Zeneca). In the ICV dapagliflozin studies, rats were injected with 4.69 µg kg$^{-1}$ dapagliflozin into a catheter in the 3$^{rd}$ ventricle of the brain (total volume: 20 µl kg$^{-1}$), or an equivalent volume of vehicle (10% DMSO/90% normal saline) by continuous infusion over a 10 min period. Four hours after the dapagliflozin infusion, a 120 min tracer infusion was initiated as described below. Unless otherwise specified ("ad lib water"), water bottles were removed at the time

of the injection. Dapagliflozin +saline treated rats were infused with normal saline (196 µL [kg-min]$^{-1}$) for six hours beginning immediately after the injection. A tracer infusion study (detailed below) was initiated at 2 p.m. After two hours of tracer infusion (4 p.m.), blood was obtained from the venous catheter and immediately placed in pre-chilled heparin-lithium tubes containing 1.25% EDTA and 1% aprotinin by volume, immediately centrifuged and plasma separated. Immediately afterward rodents were sacrificed by pentobarbital euthanasia. Rats were weighed immediately before injection of the study drugs and immediately after euthanasia. Their livers were removed and immediately freeze-clamped using tongs pre-cooled in liquid nitrogen. Urine samples were obtained directly from the bladder by collection using a syringe. Plasma and liver samples were stored at −80 ° C pending further analysis.

In the dapagliflozin +glucose infusion studies, rats were treated with dapagliflozin and a tracer study was initiated 4 h after treatment, as described above. After 1 h of tracer, blood was taken for turnover measurements and a variable infusion of glucose was initiated to raise plasma glucose concentrations to ~125 mg dL$^{-1}$ while the tracer infusion was continued. Blood was taken every 10–15 min for measurement of plasma glucose concentrations, and the glucose infusion rate was adjusted as needed. After 1 h, blood samples were taken again for turnover measurements and the rats were sacrificed as described above.

In the glucocorticoid/β−1 adrenergic blockade studies, rats were injected with dapagliflozin (1 mg kg$^{-1}$; Astra Zeneca) as described above. Two hours later, they received an IP injection of mifepristone (10 mg kg$^{-1}$ in 10% DMSO), atenolol (20 mg kg$^{-1}$ in normal saline), or both. Tracer studies were subsequently performed as described below.

To examine the impact of dehydration per se on ketoacidosis, rats were injected with furosemide (Sigma; 40 mg kg$^{-1}$) with concurrent water withdrawal and were studied 6 h later after either an 8 or a 48 h fast.

**Tracer studies.** To measure substrate turnover, rats were infused with [1,1,2,2,3–$^2$H$_5$] glycerol (2 µmol [kg-min]$^{-1}$), [$^{13}$C$_{16}$]potassium palmitate (0.5 µmol [kg-min]$^{-1}$), [1,2,3,4,5,6,6-$^2$H$_7$]glucose (1.8 µmol [kg-min]$^{-1}$), and [U-$^{13}$C$_4$]sodium β-OHB (1 µmol [kg-min]$^{-1}$) for two hours following a 5 min 3× prime. Turnover rates were determined using the equation Turnover $= \left(\frac{\text{Tracer APE}}{\text{Plasma APE}} - 1\right) \times$ Infusion rate, where APE denotes the atom percent enrichment determined by gas chromatography/mass spectrometry (GC/MS) as we have described[42]. Because the rate of β-OHB clearance (roughly calculated by multiplying the measured urine β-OHB concentration by the weight change over 6 h) was negligible (1000-fold less than the rate of β-OHB appearance, consistent with a prior study in humans[22]), β-OHB clearance was calculated by dividing the measured β-OHB turnover rate by plasma β-OHB concentrations.

**Biochemical analysis.** Plasma glucose concentrations were measured using the YSI Glucose Analyzer. Plasma and urine β-OHB and plasma bicarbonate concentrations were measured by COBAS. Plasma acetoacetate concentrations were measured using a colorimetric assay (Sigma). Plasma NEFA concentrations were measured using a Wako Diagnostics NEFA reagent, and glycerol was measured by GC/MS[42]. Plasma insulin, leptin, epinephrine, norepinephrine, corticosterone, ACTH, angiotensin II, ADH, and growth hormone concentrations were measured by ELISA (Mercodia, Abcam, Abnova, Abnova, MyBioSource, RayBiotech, MyBioSource, and Millipore, respectively). Plasma glucagon was measured by radioimmunoassay by the Yale Diabetes Research Center. Liver acetyl-CoA and malonyl-CoA concentrations were measured by LC-MS/MS[46].

**Islet culture.** Human islets from normal donors were purchased from Prodo Laboratories Inc. and were cultured immediately upon arrival in CMRL complete media consisting of glutamine free CMRL supplemented with 10 mM niacinamide and 16.7 uM zinc sulfate (Sigma), 1% ITS supplement (Corning), 5 mM sodium pyruvate, 1% Glutamax, 25 mM HEPES (American Bio), 10% HI FBS and anti-biotics (10,000 units mL$^{-1}$ penicillin and 10 mg mL$^{-1}$ streptomycin). All media components were obtained from Life Technologies unless otherwise indicated. Human donor islets were cultured as intact islets then dispersed and re-aggregated as pseudo-islets for dynamic insulin secretion studies. The formation of single pseudo-islet aggregates were performed by first dispersing the intact islets using accutase (Gibco), then the resulting cell suspension was seeded at 5000 cells per well of a 96-well V-bottom plate, lightly centrifuged (~200×g) and then incubated for 12–24 h at 37 °C 5% CO$_2$/95%.

Rat islets were isolated from Sprague-Dawley rats and recovered in culture at 37 °C 5% CO$_2$/95% for 12–24 h in 5 mM glucose RPMI (Sigma) supplemented with 10 mM HEPES (American Bio), 10% FBS (Gemini Bio-products) and antibiotics (Life Technologies). The islets were then dispersed and re-aggregated into pseudo islets as described above.

**In vitro insulin and glucagon secretion studies.** Glucose-stimulated insulin secretion studies in human and rat islets were performed 24 h after the islets were plated into the 96 well plates following dispersion and re-aggregation. The islet plates were washed and incubated at 37 °C 5% CO$_2$/95% in standard KRB (115 mM NaCl, 5 mM KCl, 24 mM NaHCO$_3$, 2.2 mM CaCl$_2$, 1 mM MgCL$_2$) supplemented with 5 mM glucose, 2 mM glutamine, 24 mM HEPES and 0.25% BSA or DMEM (Sigma D5030) supplemented with NaHCO$_3$ as per manufacturer's instructions, 5

mM glucose, 2 mM glutamine, 10 mM HEPES and 0.2% BSA for 1.5 h in presence of 10 µM dapagliflozin, 10 µM canaglifozen or DMSO (0.1% final) adjusted to the same volume for the vehicle control. The plates were then washed with glucose-free KRB or DMEM and incubated for 2 h in either KRB or DMEM study media with 0, 1, 5, 9, 11.2, 13, 16.7, or 22.4 mM glucose. Insulin and glucagon secreted from rat and human islets were measured using appropriate species ELISA plates (ALPCO and Mercodia, respectively).

**Statistical analysis.** Statistical analysis was performed using GraphPad Prism 7. Data are expressed as the mean ± S.E.M. ANOVA with Bonferroni's multiple comparisons test was used to compare three or more groups, whereas the 2-tailed Student's t-test (paired or unpaired as designated in the figure legends) was used to compare two groups.

**Reporting Summary.** Further information on experimental design is available in the Nature Research Reporting Summary linked to this Article.

## Data availability
The data generated during the current study are available from the corresponding author on reasonable request.

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

## Acknowledgements

The authors thank Jianying Dong, Xiaojian Zhao, Xian-Man Zhang, Gregori Casals, Irina Smolgovsky, Gina Butrico, Maria Batsu, and Codruta Todeasa for their valuable technical contributions and Drs. Silvio Inzucchi, Jeffrey Testani, and Hsiu-Chiung Yang for helpful discussions. This study was funded by grants from the United States Public Health Service (R01 DK-113984, P30 DK-059635, T32 DK-101019, R00 CA-215315, R01 NS-087568, UL1TR000142, T32 DK-007058), as well as an investigator-initiated award from Astra Zeneca.

## Author contributions

Data were collected and analyzed by R.J.P., A.R.C., J.D.S., R.L.C. and Y.W. The study was designed by R.J.P., J.D.S., and G.I.S. and the manuscript written by R.J.P. and G.I.S. with input from all authors.

## Additional information

**Competing interests:** G.I.S. is on the Scientific Advisory Boards for Merck, NovoNordisk, AstraZeneca, Aegerion, iMBP, Jansen Research and Development and receives investigator-initiated support from Gilead Sciences, Merck and AstraZeneca. This study was funded by an investigator-initiated award to R.J.P. and G.I.S. from Astra Zeneca. The remaining authors declare no competing interests.

