## [Peer Review File · Nature Communications]

Reviewers' comments:

Reviewer #1 (expert in SGLT2 inhibitors therapy)(Remarks to the Author):

This manuscript reports on a massive amount of experimental work carried out in the Sprague-Dawley rat model, using state-of-the-art tracer technology – to measure rates of endogenous glucose production, lipolysis, and ketogenesis – and tissue metabolite analysis. The results convincingly demonstrate that in the rat the mechanism by which SGLT2 inhibition raises endogenous glucose production, lipolysis, and ketogenesis is dual, namely, reduction in circulating insulin concentrations and increase in plasma corticosterone/catecholamine concentrations (acting in concert). Control experiments show that the latter change is secondary to volume depletion. Further experiments using intracerebroventricular injection of dapagliflozin suggest a central CNS mechanism to enhance glucagon release (rather than a direct effect of SGLT2 inhibition on pancreatic alpha cells). Finally, a separate set of studies using both rodent and human islets shows that neither dapagliflozin nor canagliflozin exert any direct effect on glucose-induced insulin secretion or glucose-mediated glucagon suppression.

Overall, the results have a high degree of internal consistency and support all the conclusions. The main concern with one of the main conclusions, namely, volume-mediated stimulation of the RAA and HPA axes and sympathetic activation, is its relevance to humans. Firstly: in normal male Sprague-Dawley rats weighing 300 g, an average ~6% placebo-corrected reduction in body weight (Fig. 2A) over just 6 hours of dapagliflozin exposure translates into a ~5 kg body weight loss in a patient with type 2 diabetes (weighing the usual 80-90 kg). This is much larger than the weight loss measured over similar time periods (7-8 hours) following acute SGLT2 inhibition (0.5-1.2 kg, refs. 22 & 23). Correspondingly, in humans urine output increases by an average ~400 mL/day and the hematocrit increases by ~3%, changes that are maintained in the long term (refs. 8, 21). Thus, it is possible that the SGLT2i-induced extracellular volume contraction is much larger in rodents than in humans, even when using a lower dose of dapagliflozin (page 6). This possibility is supported by the experiments using furosemide, which also show a large (relative to humans) reduction in body weight.

Secondly: in virtually all human studies using SGLT2i blood pressure decreases (by an average 4-6/2-3 mmHg) with no attendant increase in heart rate. In fact, in the EMPA-REG OUTCOME trial resting heart rate was decreased, instead of increased, significantly, though marginally, in the face of a reduction in blood pressure. This coupling suggests a reduction in the sympathetic/parasympathetic balance. In fact, in normotensive type 1 diabetic patients receiving empagliflozin 25 mg for 8 weeks there was no change in circulating catecholamines or heart rate variability under either euglycemic or hyperglycemic clamp conditions (Cardiovasc Diabetol 2014;13:28) despite a modest increase in circulating angiotensin II and aldosterone concentrations (Circulation 2014;129:587-97).

A final point is that the rise in β -hydroxybutyrate concentration (Fig. 2D) is far greater than what is seen in patients even with chronic SGLT2i dosing, again implying that the metabolic setting of rodents is different from man (much higher heart rate, glucose turnover (eg. Fig. 4D), fatty substrate turnover rates, etc.) and so is the response to perturbations.

It's a weakness that simple hemodynamic measures (urine volume, blood pressure, heart rate, plasma volume) were not obtained in these experiments in order to better define their equivalence (or lack thereof) to the human situation.

Reviewer #2 (expert in volume depletion)(Remarks to the Author):

This is a carefully executed study on the mechanisms of SGLT-2 inhibitor-induced euglycemic ketoacidosis. The findings are novel and provocative and have the potential to influence future thinking in the field.

Statistical analyses are properly conducted, methods are sufficiently described, the manuscript is,

with the exception mentioned below, well written and structured.

However, I feel that some of the conclusions are overreaching and the manuscript could be improved :

My major concerns are :

1. Rats: only non-obese, non-diabetic Sprague-Dawley rats were used in the experiments. Euglycemic ketoacidosis is an entity primarily observed in diabetics. The results obtained may not necessarily be extendable to diabetics where glycemia is higher and – especially in type 2 diabetes - insulinopenia not prevalent.
2. Data on volume depletion are not convincing and could be improved. Both free water access and saline administration are apparently enough to prevent weight loss and ketoacidosis during dapagliflozin treatment. This is surprising as free water is a poor ECV expander in contrast to saline. If the model proposed by the authors is true, induction of saluresis (e.g. furosemide) or aquaresis (e.g. by administration of a V2 receptor blocker) with concomitant insulinopenia (e.g. pharmacologically induced) should be sufficient to induce ketoacidosis.
3. Figure S4 : infusion of glucose to increase plasma glucose concentrations in dapagliflozin-treated rats. I would have expected a more pronounced volume depletion (and hence counterregulatory response) if filtered load is increased by exogenous glucose administration in the setting of SGLT-2 blockade.
4. Dapagliflozin-induced volume depletion is associated with a myriad of hormonal changes (increase of glucocorticoids, mineralocorticoids, Ang II, ADH, increased catecholamines). The association, however, does not necessarily indicate causality. What is the individual contribution of these pathways ? Specific pharmacologic inhibitors are available to address these questions. Results of these experiments could have direct consequences for patients treated with SGLT-2 inhibitors.
5. How do the authors explain that unilateral vagotomy is already sufficient to completely abrogate SGLT-2 inhibitor-induced hyperglucagonemia ?
6. Very high dapagliflozin doses were used, both ip and in IVC application. Dapagliflozin is a very potent inhibitor of SGLT-2 with an IC50 in the low nanomolar range. With the doses used, concomitant SGLT-1 inhibition is likely. SGLT-1 is present in central nervous system and the proximal tubule of the kidney. Even a 100 fold lower dose has been shown to induce significant glucosuria and diuresis in rats. The ip and IVC injection experiment should be repeated with a reduced, pharmacologically more relevant dose.

Minor concerns:

1. Structure of manuscript: There is no clear separation between abstract, introduction and results section. The constant switch between known and novel findings as well as facts and interpretation of data in pages 2 -5 are confusing for the reader and make it difficult to follow.

Reviewer #3 (expert in glucose metabolism) (Remarks to the Author):

In this manuscript, the authors provided the potential mechanism by which sodium-glucose transport protein 2 inhibitors (SGLT2i) provoke euglycemic ketoacidosis. They showed that SGLT2i promotes increased plasma catecholamine and corticosterone concentration in response to volume depletion, leading to the increase WAT lipolysis, hepatic glucose production and hepatic ketogenesis. In addition, insulinopenia, which is not linked to the effect by volume depletion, also affect the aforementioned phenomenon. On the other hand, SGLT2i per se cannot directly affect the glucagon secretion. Rather they showed that SGLT2i induced hepatic malonyl-CoA contents or plasma glucagon concentrations via a central mechanism, without directly contributing to the SGLT2i-induced euglycemic ketoacidosis. While the study is interesting and will be helpful to understand the molecular mechanisms behind the perplexing side effects of SGLT2i, a couple points should be addressed to strengthen the merit of the manuscript.

1. It will be helpful to format the paper based on the guidelines of the journal. The flow of the manuscript is hard to follow since the authors did not describe the figures in an orderly manner. Besides, in some cases, the description of the figures in the text and the actual figures did not match (Please refer to the last three lines of page 7, for example).

2. It will be ideal to describe the potential mechanism for the Dapagliflozin-mediated increase in plasma insulin concentration. Unlike other events such as increases in plasma epinephrine or corticosterone concentration, it doesn't appear to be induced by volume depletion.

3. Moreover, the current finding should be verified in the actual disease models such as type 1 or type 2 diabetes.

Reviewer #1

We thank Reviewer #1 for describing our manuscript as representing “a **massive amount of experimental work**...using state-of-the-art tracer technology” and are pleased that he/she believes that our “results **convincingly demonstrate** that in the rat the mechanism by which SGLT2 inhibition raises endogenous glucose production, lipolysis, and ketogenesis is dual, namely, reduction in circulating insulin concentrations and increase in plasma corticosterone/catecholamine concentrations (acting in concert).” and that “overall, the results have a **high degree of internal consistency** and **support all the conclusions**.”

Specific comments

The main concern with one of the main conclusions, namely, volume-mediated stimulation of the RAA and HPA axes and sympathetic activation, is its relevance to humans. Firstly: in normal male Sprague-Dawley rats weighing 300 g, an average ~6% placebo-corrected reduction in body weight (Fig. 2A) over just 6 hours of dapagliflozin exposure translates into a ~5 kg body weight loss in a patient with type 2 diabetes (weighing the usual 80-90 kg). This is much larger than the weight loss measured over similar time periods (7-8 hours) following acute SGLT2 inhibition (0.5-1.2 kg, refs. 22 & 23). Correspondingly, in humans urine output increases by an average ~400 mL/day and the hematocrit increases by ~3%, changes that are maintained in the long term (refs. 8, 21). Thus, it is possible that the SGLT2i-induced extracellular volume contraction is much larger in rodents than in humans, even when using a lower dose of dapagliflozin (page 6). This possibility is supported by the experiments using furosemide, which also show a large (relative to humans) reduction in body weight.

We agree that there may be differences between rat and human physiology contributing to differences in metabolism of/response to SGLT2 inhibition. Rats do have a faster metabolism than humans, and the larger weight loss with both SGLT2 inhibition and furosemide are consistent with this. However, this does not rule out relevance to humans. Of note, not every patient treated with an SGLT2 inhibitor goes into DKA. We posit that only those who do not keep up with their volume status will. Thus, while the **average** weight/volume loss in humans taking an SGLT2i may be minimal, it is likely the few patients who do become volume contracted may be the ones that are predisposed to develop euglycemic ketoacidosis.

Secondly: in virtually all human studies using SGLT2i blood pressure decreases (by an average 4-6/2-3 mmHg) with no attendant increase in heart rate. In fact, in the EMPA-REG OUTCOME trial resting heart rate was decreased, instead of increased, significantly, though marginally, in the face of a reduction in blood pressure. This coupling suggests a reduction in the sympathetic/parasympathetic balance. In fact, in normotensive type 1 diabetic patients receiving empagliflozin 25 mg for 8 weeks there was no change in circulating catecholamines or heart rate variability under either euglycemic or hyperglycemic clamp conditions (Cardiovasc Diabetol 2014;13:28) despite a modest increase in circulating angiotensin II and aldosterone concentrations (Circulation 2014;129:587-97).

As stated above, we believe that this mechanism may not occur in all patients on an SGLT2 inhibitor, hence the lack of ketoacidosis in all patients taking these agents. However, it is likely that when those on an SGLT2 inhibitor do not keep up with their volume status, this mechanism becomes relevant. This is consistent with clinical data showing that factors predisposing to DKA in SGLT2i-treated patients include “**insulin omission or dose reduction**,

severe acute illness, dehydration, extensive exercise, surgery, low-carbohydrate diets, or excessive alcohol intake” (Goldenberg et al. *Clin. Ther.* 2016). All of these factors could lead to one or both of the causative factors that we determined drive ketoacidosis in our dapagliflozin-treated rats (underline, insulinopenia; bold, dehydration). In the human studies cited by the reviewer, it is likely that the small reduction in volume (reflected in the modest observed increases in angiotensin II and aldosterone) is insufficient to produce the increases in glucocorticoids/catecholamines required to trigger euglycemic ketoacidosis in the average patient; however, the diabetic patient who is not keeping up with his/her volume status for any reason while reducing plasma insulin concentrations due to an insulin dose reduction/omission will be predisposed to develop ketoacidosis by this mechanism.

A final point is that the rise in β -hydroxybutyrate concentration (Fig. 2D) is far greater than what is seen in patients even with chronic SGLT2i dosing, again implying that the metabolic setting of rodents is different from man (much higher heart rate, glucose turnover (eg. Fig. 4D), fatty substrate turnover rates, etc.) and so is the response to perturbations.

We agree that rodents have a much higher rate of glucose and fat metabolism than humans, although they are a frequently utilized model to study the effects of novel diabetes therapies. Further studies will be required to define the relevance of these findings to the clinical scenario, but we do not believe that this invalidates the findings of this rodent study.

To address this point, we have modified the (now) penultimate sentence of the manuscript and added a final sentence to clarify that further studies will be required to test this mechanism in humans:

“Taken together this study demonstrates that two hits are required to provoke euglycemic ketoacidosis secondary to SGLT2i treatment **in the rat**, insulinopenia as well as CNS activation of the HPA axis and the sympathetic nervous system leading to increased catecholamine secretion. **Further studies will be required** to determine whether SGLT2i-induced dehydration is a potential target to prevent SGLT2i-induced euglycemic ketoacidosis and increases in hepatic glucose production in humans.”

It's a weakness that simple hemodynamic measures (urine volume, blood pressure, heart rate, plasma volume) were not obtained in these experiments in order to better define their equivalence (or lack thereof) to the human situation.

We thank Reviewer #1 for this helpful suggestion and in response to this comment we have now measured blood pressure and heart rate as requested in healthy rats (unfortunately plasma volume is practically infeasible to measure in awake rodents acutely, and we do not believe that urine volume will add significantly to these data, as it is not disputed that dapagliflozin is a diuretic). Our data demonstrate that while mean arterial pressure is not altered with dapagliflozin, likely due to the counterregulatory effects of renin-angiotensin system (which would increase blood pressure) occurring in the setting of dehydration (which would reduce blood pressure), healthy rats treated with dapagliflozin exhibited a 60% increase in heart rate (new Figs. 4A, S10C):

Next we examined these parameters in a well-established rat model of T2D treated with dapagliflozin, and similarly found that, while mean arterial pressure was unchanged, heart rate increased 75% with dapagliflozin.

These data do not indicate that the findings in this manuscript are irrelevant to the human situation. As stated above we do not believe that sympathetic activation occurs in response to dehydration in all patients on an SGLT2 inhibitor; nor do all patients on an SGLT2 inhibitor develop diabetic ketoacidosis. However, in those who do, our data indicate that it is likely that both dehydration leading to increased plasma catecholamines/glucocorticoids and insulinopenia are key predisposing factors.

Reviewer #2

We thank Reviewer #2 for his/her many positive comments, describing this as “a **carefully executed** study on the mechanisms of SGLT-2 inhibitor-induced euglycemic ketoacidosis,” presenting “findings [which] are **novel and provocative** and have the **potential to influence future thinking** in the field.” We further thank the reviewer for commenting that the “statistical analyses are **properly conducted**, methods are **sufficiently described**, the manuscript is, with the exception mentioned below, **well written and structured**.”

Specific comments

- Rats: only non-obese, non-diabetic Sprague-Dawley rats were used in the experiments. Euglycemic ketoacidosis is an entity primarily observed in diabetics. The results obtained may not necessarily be extendable to diabetics where glycemia is higher and – especially in type 2 diabetes - insulinopenia not prevalent.**

In response to this comment, we have repeated the dapagliflozin±*ad lib* water treatment studies in rats treated with low-dose streptozotocin and nicotinamide, a well-established rodent model of type 2 diabetes. We find that dapagliflozin lowers plasma glucose and insulin concentrations in this model, without generating absolute insulinopenia:

As in control animals, dapagliflozin led to ketoacidosis, increasing rates of ketone turnover and plasma β-OHB concentrations while lowering plasma bicarbonate, demonstrating diabetic ketoacidosis:

Also similar to control rats, ketosis in T2D rats was driven by increased rates of white adipose tissue lipolysis as reflected by increased fatty acid concentrations and turnover, as well as increases in glycerol turnover, although each parameter was increased at baseline as compared to non-diabetic rats:

These increased rates of white adipose tissue lipolysis in turn increased hepatic acetyl-CoA content and endogenous glucose production.

The observed increases in rates of lipolysis, ketogenesis, and endogenous glucose production were driven by with dehydration, with T2D rats manifesting an 8% body weight loss following treatment with dapagliflozin if not given free access to water.

However, preventing volume depletion by giving dapagliflozin-treated rats free access to drinking water prevented dapagliflozin-induced increases in weight loss, white adipose tissue lipolysis, and ketogenesis. To evaluate the physiologic link between dehydration and ketoacidosis, we measured corticosterone and glucocorticoid concentrations and found that dehydration was associated with increases in concentrations of each hormone,

as well as with tachycardia reflecting the observed sympathetic activation.

Finally, similar to humans treated with SGLT2 inhibitors and control rats in the current study treated with dapagliflozin, T2D rats exhibited hyperglucagonemia beyond the inappropriate hyperglucagonemia already observed in the T2D control animals:

However, our data dissociating hyperglucagonemia from ketoacidosis in ICV dapagliflozin-treated rats (which exhibit hyperglucagonemia without ketoacidosis) and vagotomized, dapagliflozin-treated rats (which exhibit ketoacidosis without hyperglucagonemia) indicate that the hyperglucagonemia observed in these T2D rats likely has a minimal effect to drive dapagliflozin-induced ketoacidosis in T2D rodents.

2. Data on volume depletion are not convincing and could be improved. Both free water access and saline administration are apparently enough to prevent weight loss and ketoacidosis during dapagliflozin treatment. This is surprising as free water is a poor ECV expander in contrast to saline. If the model proposed by the authors is true, induction of saluresis (e.g. furosemide) or aquaresis (e.g. by administration of a V2 receptor blocker) with concomitant insulinopenia (e.g. pharmacologically induced) should be sufficient to induce ketoacidosis.

To address this point, we induced insulinopenia with a 48 hr fast and treated with furosemide.

This method of inducing insulinopenia is preferable to the use of pharmacologic agents because the use of such agents to completely abrogate the ability of the β-cell to secrete insulin will place them in pharmacologically induced diabetic ketoacidosis, a state in which it would be difficult to detect dapagliflozin-induced ketoacidosis beyond the baseline.

However, and consistent with our previous data (Perry et al. *Cell* 2018), 48 hr fasted Sprague-Dawley rats were severely insulinopenic (~20 pM). In this setting, furosemide again caused dehydration resulting in increases in plasma corticosterone and catecholamine concentrations leading to increases in heart rate and rectal temperature reflecting the observed sympathetic activation:

And in the setting of hyperinsulinemia induced by a prolonged fast, furosemide treatment did increase white adipose tissue lipolysis and ketoacidosis, with rates of fatty acid and glycerol turnover increasing by ~70%, and β -OHB turnover increasing by 60%, resulting in ketoacidosis as reflected by reductions in bicarbonate concentrations:

In addition, the increased lipolysis observed with furosemide treatment in the setting of insulinopenia increased hepatic acetyl-CoA content, thereby increasing hepatic glucose production likely as a result of activation of pyruvate carboxylase (Perry et al. *Cell* 2015), and modestly increasing plasma insulin concentrations:

However, plasma glucagon concentrations were unchanged with furosemide treatment in 48 hr fasted rats, providing another model in which ketoacidosis due to the combination of volume depletion and insulinopenia was dissociated from plasma glucagon concentrations.

3. Figure S4: infusion of glucose to increase plasma glucose concentrations in dapagliflozin-treated rats. I would have expected a more pronounced volume depletion (and hence counterregulatory response) if filtered load is increased by exogenous glucose administration in the setting of SGLT-2 blockade.

While we agree this could be a possible outcome we clearly did not observe any counterregulatory response as we did in our other perturbations.

4. Dapagliflozin-induced volume depletion is associated with a myriad of hormonal changes (increase of glucocorticoids, mineralocorticoids, Ang II, ADH, increased catecholamines). The association, however, does not necessarily indicate causality. What is the individual contribution of these pathways? Specific pharmacologic inhibitors are available to address these questions. Results of these experiments could have direct consequences for patients treated with SGLT-2 inhibitors.

As Reviewer #2 requests, we have now undertaken a large number of studies using pharmacologic inhibitors of glucocorticoid and catecholamine (β -1 adrenergic) activity to determine the perturbations responsible for the observed increases in lipolysis and ketoacidosis with SGLT2 inhibitor treatment. Our data demonstrate that β -1-adrenergic and glucocorticoid activity have additive effects to promote lipolysis and ketoacidosis.

In addition, both β -1-adrenergic and glucocorticoid activity contribute to dapagliflozin's effect to increase hepatic acetyl-CoA content and endogenous glucose production, with either sufficient to generate most of the increase in either parameter.

Of note, blocking both β -1-adrenergic and glucocorticoid receptor activity fully abrogated any effect of dapagliflozin to promote ketoacidosis, demonstrating that increased plasma catecholamine and corticosterone concentrations are necessary for dapagliflozin-induced ketoacidosis and dissociating any possible alterations in mineralocorticoids, angiotensin II, and ADH from ketoacidosis resulting from dapagliflozin treatment.

5. How do the authors explain that unilateral vagotomy is already sufficient to completely abrogate SGLT-2 inhibitor-induced hyperglucagonemia?

The majority of the parasympathetic innervation to the pancreas comes through the left vagus (Bergman and Miller, *AJP* 1973; Ahrén and Taborsky, *Endocrinology* 1986; Liu et al. *J. Himforsch.* 1998; Rozman and Bunc, *Exp. Physiol.* 2004; Babic and Travagli 2016, DOI: [10.3998/panc.2016.27](https://doi.org/10.3998/panc.2016.27)), so it is not surprising that a left-sided vagotomy would abrogate most of the effect of dapagliflozin to promote hyperglucagonemia. In addition, it is important to note that dapagliflozin does still increase plasma glucagon three-fold ($p=0.006$) in unilateral vagotomized rats:

Therefore, it is not accurate to state that unilateral vagotomy completely abrogates SGLT2 inhibitor-induced hyperglucagonemia. Our data are consistent with the aforementioned

reports indicating that the left vagus provides the majority, but not all, of the parasympathetic input to the pancreas.

6. Very high dapagliflozin doses were used, both ip and in IVC application. Dapagliflozin is a very potent inhibitor of SGLT-2 with an IC₅₀ in the low nanomolar range. With the doses used, concomitant SGLT-1 inhibition is likely. SGLT-1 is present in central nervous system and the proximal tubule of the kidney. Even a 100 fold lower dose has been shown to induce significant glucosuria and diuresis in rats. The ip and IVC injection experiment should be repeated with a reduced, pharmacologically more relevant dose.

To address this concern, we have repeated these studies with an intermediate dose of dapagliflozin (1 mg kg⁻¹). This lower dose was chosen to provide pharmacokinetics more similar to the typical human dose (10 mg in a ~70 kg patient) while accounting for the approximately ten-fold faster metabolism seen in rodents as compared to humans. Our data in type 2 diabetic rats treated with 1 mg kg⁻¹ dapagliflozin replicate our data in rats treated with the higher (10 mg kg⁻¹) dose of the drug. In a rat model of type 2 diabetes, dapagliflozin caused dehydration, resulting in increases in plasma catecholamine and corticosterone concentrations and reflected in a 75% increase in heart rate.

These increases in lipolytic hormone concentrations increased white adipose tissue lipolysis,

hepatic acetyl-CoA and endogenous glucose production.

Together, these perturbations resulted in diabetic ketoacidosis indicated by >2-fold increases in plasma β -OHB concentrations and whole-body β -OHB turnover, associated with a 25% reduction in plasma bicarbonate concentrations:

Thus, taken together these data demonstrate that an intermediate, pharmacologically relevant dose of dapagliflozin replicates the data obtained with higher doses of the drug. However, given that the phenotype in ICV dapagliflozin injected rats was minor, with alterations seen only in plasma glucagon concentrations, but this perturbation dissociated from ketogenesis, we do not believe that much would be gained by repeating this largely negative study with another ICV dapagliflozin dose.

Minor concerns

- 1. Structure of manuscript: There is no clear separation between abstract, introduction and results section. The constant switch between known and novel findings as well as facts and interpretation of data in pages 2 -5 are confusing for the reader and make it difficult to follow.**

We apologize for these formatting issues, some of which arose because of the direct transfer of this Letter from *Nature Medicine* to *Nature Communications*, and have reformatted as requested.

Reviewer #3

We thank the reviewer for describing this study as “**interesting and will be helpful**” to understand the molecular mechanisms behind the perplexing side effects of SGLT2i.”

1. It will be helpful to format the paper based on the guidelines of the journal. The flow of the manuscript is hard to follow since the authors did not describe the figures in an orderly manner. Besides, in some cases, the description of the figures in the text and the actual figures did not match (Please refer to the last three lines of page 7, for example).

We apologize for any formatting errors and have undertaken an extensive reformatting of the manuscript as suggested by the reviewer.

2. It will be ideal to describe the potential mechanism for the Dapagliflozin-mediated increase in plasma insulin concentration. Unlike other events such as increases in plasma epinephrine or corticosterone concentration, it doesn't appear to be induced by volume depletion.

This is an error – plasma insulin actually decreases by 60% with dapagliflozin treatment:

This reduction in plasma insulin occurs because of the dapagliflozin-induced lowering of plasma glucose concentrations, as shown when we infuse glucose in dapagliflozin treated rats (and insulin consequently increases).

3. Moreover, the current finding should be verified in the actual disease models such as type 1 or type 2 diabetes.

As requested by the reviewer, we have now treated a well-established rat model of type 2 diabetes with dapagliflozin ± volume replacement (*ad lib* access to water). Our data replicate the key findings observed in healthy rats treated with dapagliflozin. In T2D animals, dapagliflozin caused dehydration, demonstrated by an 8% body weight loss, associated with 200-350% increases in plasma ADH and angiotensin II concentrations:

This dehydration resulted in increases in sympathetic and glucocorticoid activity,

which increased white adipose tissue lipolysis.

Increased substrate delivery to the liver resulted in increases in hepatic acetyl-CoA concentrations and ketone production,

doubling endogenous glucose production and causing diabetic ketoacidosis:

This increase in endogenous glucose production with dapagliflozin treatment was associated with a doubling in plasma glucagon concentrations;

however, we present extensive data in this manuscript **dissociating** hyperglucagonemia from ketogenesis. First, we show that an ICV injection of dapagliflozin, at a dose that does not have systemic effects as evidenced by the absence of glycosuria, causes hyperglucagonemia without ketoacidosis:

Next, we find that a unilateral vagotomy abrogates, to a large extent, the hyperglucagonemia induced by dapagliflozin treatment. However, again in vagotomized rats, plasma glucagon concentrations are dissociated from ketoacidosis: vagotomized rats exhibit ketoacidosis without hyperglucagonemia:

Taken together these data demonstrate that whereas dapagliflozin causes hyperglucagonemia in both healthy and T2D rats, this hyperglucagonemia is neither necessary nor sufficient to drive ketoacidosis.

REVIEWERS' COMMENTS:

Reviewer #1 (Remarks to the Author):

The argument that what is seen in this rodent model applies (and explains) what is seen in so-called euglycemic DKA in patients with type 1 or type 2 diabetes on SGLT2i treatment (a fraction of a percent of users) is weak: insulin lack, carbohydrate deficit and dehydration are common mechanisms of progression towards DKA, which SGLT2 inhibition may exacerbate but definitely mask because of the modest degree of hyperglycemia.

The various rat models in this study document the effects of extreme, acute dehydration (with accelerated heart rate and sympathetic activation) rather than the pharmacological effect of dapagliflozin per se. This interpretation is supported by the furosemide experiments.

Reviewer #2 (Remarks to the Author):

The authors did an excellent job in addressing all my concerns.

Reviewer #3 (Remarks to the Author):

The authors revised the manuscript adequately by addressing all the issues raised by the reviewers. I have no further comments.

Referee #1

The argument that what is seen in this rodent model applies (and explains) what is seen in so-called euglycemic DKA in patients with type 1 or type 2 diabetes on SGLT2i treatment (a fraction of a percent of users) is weak: insulin lack, carbohydrate deficit and dehydration are common mechanisms of progression towards DKA, which SGLT2 inhibition may exacerbate but definitely mask because of the modest degree of hyperglycemia.

The various rat models in this study document the effects of extreme, acute dehydration (with accelerated heart rate and sympathetic activation) rather than the pharmacological effect of dapagliflozin per se. This interpretation is supported by the furosemide experiments.

We agree that not all individuals on dapagliflozin develop dehydration – or ketoacidosis. Indeed ketoacidosis is fairly rare; however it is still important to understand why, in rare circumstances, ketoacidosis can develop. This study demonstrates that the combination of dehydration and insulinopenia – which may be rare but do occur – predisposes to ketoacidosis, at least in rats. That said, we have added several comments highlighting that these studies were performed in rats and that “Further studies will be required to determine whether these findings in healthy and diabetic rodents will translate to humans, who will typically – but not always – remain hydrated during SGLT2i treatment.” Finally we point the referee’s attention to the final sentence of the manuscript, which specifically addresses this point: “Further studies will be required to determine whether SGLT2i-induced dehydration is a potential target to prevent SGLT2i-induced euglycemic ketoacidosis and increases in hepatic glucose production in humans.”